# Outer Membrane Vesicles (OMVs) as Biomedical Tools and Their Relevance as Immune-Modulating Agents against *H. pylori* Infections: Current Status and Future Prospects

**DOI:** 10.3390/ijms24108542

**Published:** 2023-05-10

**Authors:** Abeer Ahmed Qaed Ahmed, Roberta Besio, Lin Xiao, Antonella Forlino

**Affiliations:** 1Department of Molecular Medicine, Biochemistry Unit, University of Pavia, 27100 Pavia, Italy; abeerahmedqaed.ahmed@unipv.it (A.A.Q.A.); roberta.besio@unipv.it (R.B.); 2School of Biomedical Engineering, Shenzhen Campus, Sun Yat-sen University, Shenzhen 518107, China; xiaolin23@mail.sysu.edu.cn

**Keywords:** outer membrane vesicles, *Helicobacter pylori*, immune modulation, biomedical applications

## Abstract

Outer membrane vesicles (OMVs) are lipid-membrane-bounded nanoparticles that are released from Gram-negative bacteria via vesiculation of the outer membrane. They have vital roles in different biological processes and recently, they have received increasing attention as possible candidates for a broad variety of biomedical applications. In particular, OMVs have several characteristics that enable them to be promising candidates for immune modulation against pathogens, such as their ability to induce the host immune responses given their resemblance to the parental bacterial cell. *Helicobacter pylori* (*H. pylori*) is a common Gram-negative bacterium that infects half of the world’s population and causes several gastrointestinal diseases such as peptic ulcer, gastritis, gastric lymphoma, and gastric carcinoma. The current *H. pylori* treatment/prevention regimens are poorly effective and have limited success. This review explores the current status and future prospects of OMVs in biomedicine with a special focus on their use as a potential candidate in immune modulation against *H. pylori* and its associated diseases. The emerging strategies that can be used to design OMVs as viable immunogenic candidates are discussed.

## 1. Introduction

Bacterial membrane vesicles, originating from both Gram-negative and Gram-positive bacteria, play various roles in bacterial survival and biological functions [1,2,3,4], including microbial virulence, cellular crosstalk, and host immune response modulation. Furthermore, they possess unique targeting and packaging abilities [5,6,7]. Bacterial membrane vesicles derived from Gram-positive bacteria are called membrane vesicles (MVs) as they are originating from the cytoplasmic membrane, while bacterial membrane vesicles derived from Gram-negative bacteria are called outer membrane vesicles (OMVs) as they originate from the outer membrane of the bacterial cell (Figure 1). They both range between 20 and 500 nm in size and contain parental bacterial cell materials [3,8,9,10,11,12], but since they originate from different parts, their contents vary accordingly. For instance, the surface of OMVs contains the same components of the outer membrane (Figure 1A), while the surface of MVs contains the same components of the cytoplasmic membrane (Figure 1B).

## 2. OMVs

OMVs are lipid-membrane-bounded nanoparticles that are secreted via outer membrane vesiculation of Gram-negative bacteria to contribute to different biological processes [7,13]. Even if OMVs were observed in early reports, they did not receive any attention and their importance had been overlooked until they were found in the spinal fluid of patients infected with meningitis [14]. Since then, the understanding of OMVs’ biogenesis, function, production, and how they contribute to the interaction between the bacterial cell and the host has received increasing attention.

OMVs are composed of several components such as lipids, proteins (e.g., enzymes and structural proteins), carbohydrates, and genetic material (DNA and RNAs), which they originally inherited from the parental cells (Figure 1A) [3]. In the biomedical field, OMVs play essential roles in attenuating and treating diseases [15,16]. For instance, OMVs can play a significant anti-infection role by inducing and modulating immune responses or by inhibiting pathogen localization and proliferation. Thus, OMVs are recognized as promising candidates for various biomedical applications such as immune modulation, drug delivery, cancer therapy, vaccine development, and anti-bacterial treatments [15,17,18,19,20,21,22,23,24,25,26]; however, their full potential, advantages, future perspectives, and associated problems need to be further investigated.

### 2.1. OMV Biogenesis

Several models, briefly summarized below, have been reported to explain the vesiculation mechanism (Figure 2, Appendix A) [3]. In all cases, vesiculation allowed the separation of the outer membrane from the below peptidoglycan layer and budding outward until a vesicle can form and separate from the bacterial cell surface. However, one exception was reported, which described the vesicle formation mechanism by the “explosive” cell lysis that is initiated via a prophage endolysin [27].

One of the currently reported models for OMV biogenesis is vesiculation via the VacJ/Yrb ATP-binding cassette ABC transporter, belonging to the family of phospholipid transporters (Figure 2A). This mechanism is associated with the accumulation of phospholipids in the outer membrane’s outer leaflet due to the transcriptional silencing or inactivation of the VacJ/Yrb transporter, which is responsible for the maintenance of outer membrane lipid asymmetry [28,29]. This trafficking system is highly conserved in Gram-negative bacteria and was primarily reported to be responsible for phospholipid transportation from the outer membrane to the inner cytoplasmic membrane [30]. The accumulation of phospholipids inside the outer membrane leaflet due to the downregulation of the Vac/Yrb transporter induces the outward curvature which facilitates the formation of the outer membrane budding and OMV release (Figure 2A).

The second model of OMV biogenesis involves the insertion of some molecules into the outer membrane outer leaflet (e.g., B-band of lipopolysaccharides (LPS) and *Pseudomonas aeruginosa* (*P. aeruginosa*) quinolone signal (PQS)), which can trigger the outward bulging of the outer membrane and promote OMV formation (Figure 2B). For instance, the B-band of LPS was proposed to localize to a specific area of the outer membrane, and because of the close proximity of similar charges, it was hypothesized to force the outer membrane to bulge out [31]. Similarly, the insertion of PQS into the outer membrane’s outer leaflet was found to increase the OMV production in Gram-negative bacteria [32]. PQS interacts with lipid A of LPS and sequesters cations such as Ca^2+^ and Mg^2+^. The anionic repulsion that occurs between neighboring LPSs could lead to outer membrane blebbing and OMV formation. Moreover, the interaction of PQS with lipid A decreases LPS fluidity, facilitating the outer leaflet expansion and promoting curvature [33]. The PQS-based model is considered one of the best-investigated models. However, it is species-specific since PQS is only produced by *P. aeruginosa*. 

The third model is based on the presence of specific types of LPS, and/or phospholipids that are enriched in the outer membrane areas where vesiculation occurs (Figure 2C) [34]. These molecules have the ability to induce vesiculation due to their charges or atypical structure that promote the outward bulging of the outer membrane, and consequently the release of the OMVs into the external milieu. The elevated levels of the negatively charged LPS in certain areas of the outer membrane are an indication of the induction of OMV formation in these areas that could be triggered by specific conditions such as oxidative stress (Figure 2C) [35,36]. For instance, OMVs isolated from *P. aeruginosa* were found to primarily consist of negatively charged LPS.

The fourth model associates the accumulation of misfolded proteins, peptidoglycan fragments, and other molecules inside the periplasmic space with the increase in local pressure on the outer membrane responsible for OMV formation (Figure 2D) [37,38]. It was proposed that vesiculation occurs as a protective mechanism to remove toxic and/or unwanted cellular components. Indeed, vesiculation was increased in *Escherichia coli* (*E. coli*) carrying a deletion of DegP, which is a periplasmic chaperone/protease known to correlate with the stress response in the envelope. DegP activity prevents the accumulation of misfolded or damaged proteins inside the periplasm [38]. The role of the periplasmic chaperone/protease in avoiding the accumulation of toxic components was identified in several Gram-negative bacteria, which validates this model for OMV biogenesis (Figure 2D) [38,39,40].

The fifth model proposes the disruption of crosslinks between the peptidoglycan layer and lipoproteins as a determining step for OMV formation (Figure 2E). The outer membrane of Gram-negative bacteria is well known to be stabilized by the crosslinks between lipoproteins present in the outer membrane and the underlying peptidoglycan layer located in the periplasmic space. However, the lack of these crosslinks in some areas of the outer membrane allows the outer membrane in these regions to curve and form OMVs [41]. Thus, the biogenesis of OMVs is maintained by specific enzymes that are involved in the outer membrane–peptidoglycan layer interactions [42], such as enzymes involved in peptidoglycan synthesis and breakdown (e.g., peptidoglycan endopeptidases) [43]. 

Despite all the efforts to describe the OMV biogenesis, more investigations are required to better understand the process and to investigate why certain molecules/components are present in the OMVs. For instance, the current OMV biogenesis models can explain the presence of some OMV contents (e.g., phospholipids, LPS, and peptidoglycan) but cannot explain the presence of others such as DNA and other bacterial contents that are usually present in the cytoplasm. 

### 2.2. OMVs in Biomedical Applications

One of the strengths of using OMVs in biomedicine is the possibility of using them as therapeutic vehicles. Various factors should be considered in developing extracellular vesicles for biomedical applications; e.g., they should be cost-effective, easy to synthesize, biocompatible, non-toxic, feasible to scale up, and with high therapeutic efficacy [44]. Other extracellular vesicles such as exosomes lack important aspects of their potentiality such as the unfeasibility of undertaking large-scale mammalian cultures for vesicle production. OMVs have been suggested as a viable alternative that can be manufactured and produced easily at a large scale and at lower cost [44]. Moreover, OMVs can combine several desirable effects such as delivering the targeted drug (e.g., chemotherapeutic agents) into the specific (e.g., tumor) microenvironment and at the same time recruit immune cells into it, and therefore, enhance their efficacy with no apparent toxicity [45,46,47]. In this regard, naturally acquired OMVs could be used directly or could be modified or genetically engineered to achieve this aim [20,24,25,48,49]. Naturally derived OMVs can serve as DNA, RNA, antigen, and antibody carriers, or as delivery vehicles for their natural cargos (Figure 3A). Meanwhile, the modified OMVs can be used in various applications depending on the needed function: they can serve as carriers of specific nanoparticles, DNA and RNA molecules, antigens, antibodies, or drugs (Figure 3A). The modifications of OMVs can be applied by loading the desired cargos inside the OMV lumen, by warping nanoparticles inside the OMVs, by concealing OMVs inside nanoparticles, or by embedding the desired components (e.g., antigen, antibody, ligand, etc.) within the outer membrane layer (Figure 3B). Cargo loading into OMVs can be performed using different techniques, one of these being electroporation, which involves the use of high-voltage pulses to create pores in the membrane of OMVs, which leads to a temporary permeable state [50,51,52]. This temporary permeability allows the loading of drugs, proteins, nucleotides, small-sized nanoparticles (e.g., metallic gold nanoparticles, AuNPs), etc., which can be achieved using different electric pulses at different durations. After loading the desired molecules, the membrane of OMVs can recover its original structure and lose the temporary permeability without any damage. Similarly, the treatment of OMVs with saponin containing reagents increases their membrane permeability, which facilitates cargo loading without damaging the membrane structure [53,54]. The controlled and temporary disruption of OMV membranes allows cargo loading to also be achieved by applying multiple freeze–thaw cycles in a buffer that contains the material of interest [55,56,57]. 

The co-extrusion technique is a process of repeating mechanical extrusion using polycarbonate filter membranes that have various pore sizes, which allows the loading of the desired cargo into the OMVs [58]. In this method, OMVs are mixed with the cargo of interest (e.g., drugs, nanoparticles, etc.) and extruded together to force them to interact [59]. Similarly, sonication can be applied as a simpler method for OMV loading. Ultrasonic frequencies can be applied to a mixture of OMVs with the material of interest. This leads to their loading or could result in the attachment of the cargo to the surface of the OMVs due to the temporary disruption of their membrane [60,61]. On the other hand, a simpler alternative technique such as incubating OMVs with the material of interest can be applied. For example, OMVs from *Klebsiella pneumoniae* (*K. pneumoniae*) were loaded with doxorubicin hydrochloride (chemotherapeutic drug) by incubating the drug with the OMVs at 37 °C for 4 h [47]. Similarly, the loading process can also be applied by incubating the bacteria of interest with the cargo material during the bacterial growth phase. In this method, the bacteria engulf the material of interest that is present in the medium, pack it into the OMVs, and then release it into the extracellular medium. A study by Huang, et al. [62] successfully used this method to synthesize antibiotic-loaded OMVs from *Acinetobacter baumannii* (*A. baumanii*) that resulted in effectively killing certain bacteria in vitro and in vivo [62]. 

Genetic engineering could be applied to add certain molecules to the surface of OMVs [48,49]. Cargo loading can be applied by the transformation of bacteria using an engineered plasmid that expresses the desired cargo [63,64]. Using this method, the material of interest such as antibodies, antigens, enzymes, and proteins can be loaded into the OMVs [63,64,65,66,67]. Genetic engineering techniques allow the use of different methods to load various types of cargo into OMVs. For instance, recombinant DNA technology enables introducing specific modification into OMVs that can be beneficial for a specific desirable application (e.g., inserting antigens for immune modulation). In addition, genetic engineering can also be applied to knockout genes responsible for a specific undesirable function to be eliminated from the generated OMVs, such as knocking out genes responsible for toxic proteins [68,69]. 

In short, natural or modified/engineered OMVs could serve as nanopharmaceuticals based on their desired characteristics in a variety of biomedical applications, such as vaccines, adjuvants, cancer immunotherapy, drug delivery, and anti-bacterial adhesion agents (Figure 3) [15,70,71,72,73,74,75,76,77,78]. 

OMVs are considered excellent vaccine candidates against pathogenic bacteria, and can be used as antigens to induce cellular (cytokines and activated T cells) and humoral (antibody) immune responses after immunization of humans and animals (Figure 4A) [17,18]. The first vaccine trial of OMVs was in 1991 and was employed against *Neisseria meningitidis* (*N. meningitidis*) [79]. Meningitis type B (MenB) is an OMV-based vaccine that is currently approved to treat patients [80,81,82]. Subsequently, efforts continued for new vaccine development against various diseases caused by pathogenic Gram-negative bacteria [49,83,84]. However, there are no other OMV-based vaccines to treat pathogenic Gram-negative bacteria currently available on the market. 

OMVs can also be employed as adjuvants to enhance the immune responses against an antigen (Figure 4B). This can be achieved by mixing the OMVs with the antigen of interest in the vaccine preparations, linking the antigen to the OMV surfaces, loading the antigen inside the OMVs, or by genetically engineering the bacteria to express the antigen in their outer membrane and consequently into their released OMVs [15]. After immunization, the OMV-containing formulations could trigger robust cellular and humoral immune responses. Contrary to most of the other classic adjuvants that cause systemic and local hypersensitivity, OMVs were found to have low toxicity as well as high potency for inducing T cell responses [85]. 

The use as agents in cancer immunotherapy to annihilate tumor tissues is another intriguing OMV application (Figure 4C). OMVs have been proposed as a good platform for anti-tumor vaccine development for several reasons, such as OMV strong immunogenicity, the ability of OMVs to carry the anti-tumor antigen (inside the vesicle or on its surface), to enhance the antigen presentation, and the lack of OMV proliferation [19,20]. OMV-based anti-tumor vaccines are primarily developed by genetic engineering to express a foreign protein inside the OMVs or linked to the OMV surfaces. This antigen should have the ability to induce the required immune response against cancer cells without causing undesired side effects. OMV-based anti-tumor vaccines can be used to kill cancer cells and/or to silence relevant genes [20]. Various bacterial components such as enzymes, peptides, and toxins have been investigated for cancer therapy [86]. OMVs provide a unique vehicle to combine several anti-tumor components that can initiate an immune response, which is considered a sought-after cancer immunotherapy agent. For instance, OMVs contain parental components (e.g., LPS) that can stimulate an immune response that enables immune cell maturation and tumor damage [45,46]. Moreover, OMVs can function as nanocarriers for loading chemotherapeutic agents. Indeed, the use of doxorubicin passive-loaded OMVs isolated from the attenuated *K. pneumonia* not only caused a cytotoxic effect and cell apoptosis resulting from the doxorubicin, but it was also observed in vivo that OMVs worked synergistically with their cargo to recruit macrophages into the tumor microenvironment, which enhanced the anti-tumor efficacy with no apparent toxicity [47]. 

OMVs are also considered an efficient delivery system to transport their cargo to other cells and/or microenvironments (Figure 4D). OMV contents can be transported to any targeted cell through two possible mechanisms. The first system proposes the spontaneous lysing of OMVs, which allows their contents to diffuse. The second mechanism is based on the OMV fusion with the targeted cell, on their proximal lysis or internalization [21]. OMV ability for long-distance transportation is one of the main strengths of the OMVs as a vehicle for drug delivery. OMVs can enhance the pharmacodynamics and pharmacokinetics of the loaded drugs by extending the blood circulation time as well as by protecting the loaded molecules from degradation [66]. OMVs have an outstanding targeting capability for bacteria, cells, or inflammatory sites through surface functional protein modifications [24]. Genetic engineering can be applied to express specific targeting ligands onto OMV surfaces [48]. Moreover, pathogen-associated molecular patterns (PAMPs) on the OMVs facilitate their recognition and ingestion by immune cells, which hold great potential for targeted drug delivery against immune cells [87]. OMVs inherit the same surface antigens as the parental cell. Thus, they have the ability to be ingested and recognized by the immune cell, and this can be beneficial in targeted therapy. 

The identification of pathogenic bacteria in humans is sometimes difficult for several reasons such as localized infections or aggressive antibiotics concurrent treatments as well as slow-growing bacteria and poor sensitivity of the diagnostic methods [22]. For instance, the analysis of more than 2.5 million sepsis cases using the Premier Healthcare Database in the United States showed that the specific causal organism responsible for sepsis could not be identified in over 70% of the cases [88]. In another study by Stranieri et al. [89], the causal organism of neonatal sepsis was identified only in 41% of the blood cultures from patients [89]. Indeed, bacterial cultures can take up to 24 h to grow and thus they are not compatible with a quick diagnosis and proper specific antimicrobial treatment that is suggested within 1 to 3 h from the recognition of the infection types (e.g., sepsis) [90]. While bacterial cultures fail to provide an accurate and fast method to identify the bacteria that caused the infection, OMVs can persist and permit a more definitive diagnostic approach [91,92]. Due to OMV size, they can widely circulate in the body and freely cross tissue barriers, which allows efficient diagnosis from easily obtained biofluids (e.g., urine or blood) [22]. As stated above, OMVs contain various components of the parental cells. The OMV cargo conserved among bacteria can act as ideal biomarkers for their presence; meanwhile, the OMV cargo specific for a given bacterium can be extremely useful as a rapid differentiation and identification tool for bacterial species identification in OMVs isolated from biofluids (Figure 3E) [22]. For instance, the widely expressed LPS can serve as a biomarker for Gram-negative bacteria, whereas a species-specific component (e.g., 16S r RNA, urease A (UreA) and heat shock protein (Hsp60) for *H. pylori*) that is conserved within the targeted bacterial species can be used as a biomarker for the bacteria of interest following its characterization [93,94,95,96]. The presence among pathogenic bacteria of species-specific repeats, both at genomic and protein levels, can be identified using computational methods [97,98,99]. This allows the recognition of highly conserved species-specific hallmarks. Subsequently, they can be used as reliable biomarkers for identifying a specific species present in biological fluids that contained the targeted bacteria or its OMVs. 

OMVs also play a vital role in cell–cell communication since signaling molecules can be protected inside the OMV lumen until they reach the target location (Figure 4F). For instance, it was reported that 86% of total PQS were packed inside the OMV-derived *P. aeruginosa* [23]. When these OMVs were removed from the bacterial population, the PQS-controlled group behavior and cell–cell communication were inhibited. Similarly, the hydrophobic quorum-sensing molecule CAI-1 and the hydrophobic signal N-hexadecanoyl-L-homoserine lactone from *Vibrio harveyi* and *Paracoccus* sp., respectively, which are responsible for coordinating bacterial group behavior and involved in long-distance communication, were found to be present inside the OMVs [100,101]. OMVs spread far from their parental cell, and therefore, they can be considered as an intra-kingdom communication mechanism that enables the transportation of signaling molecules [102,103]. In addition, OMVs can facilitate trans-kingdom exchange and the delivery of biomolecules between bacteria and their hosts [104]. 

OMVs are a secretory system, as they have the ability to disseminate bacterial products to their environment (Figure 4G), but with unique features. Besides the cargo protection described above, unlike other systems, OMV-mediated secretion can allow the simultaneous secretion of various soluble and insoluble compounds such as membrane proteins, lipids, and insoluble molecules [31,105,106]. Furthermore, the OMV-mediated secreted materials can be delivered and transported at high concentrations, which is often needed for proper efficacy [31,107], and a specifically targeted delivery of molecules can be obtained via selective binding between surface bacterial adhesins and the receiver’s receptors and ligands [31,108]. The selective transportation of OMV cargo to other bacterial cells was observed both in the same or different species. The targeting abilities of OMVs to specific cells hold great potential for the targeted delivery of molecules. 

OMVs are mimics of their parental bacteria, and thus they have the ability to inhibit the adhesion of their parental pathogenic bacteria onto the host cell by competitively binding to the same target site [15,24]. Bacterial infections are initiated by bacterial cell adhesion to the targeted cell, and therefore, the anti-adhesion treatment can offer a promising therapy in comparison to other conventional therapies that might induce antibiotic resistance [109]. In addition, the combination of antibiotic and anti-adhesion therapies shows collaborative antibacterial efficacy [110]. OMV-derived *H. pylori* were reported to block *H. pylori* adhesion to gastric epithelial cells by developing OMV-coated nanoparticles that preserved the *H. pylori* surface antigens [24]. Moreover, these developed OMV nanoparticles reduced *H. pylori* attachment in mouse stomach tissues, which indicates OMV potential in the inhibition of bacterial adhesion to the host tissues. OMV anti-adhesion efficacy can be further improved through genetic engineering to regulate adhesion expression as well as by choosing the ideal nanoparticle cores with desired properties [111,112]. In addition, the use of OMVs as anti-bacterial adhesion tools can promote bacterial clearance, uptake, and recognition by immune cells [15,113]. Overall, OMVs can be considered an efficient anti-adhesion intervention to fight bacterial infections (Figure 4H). 

In recent years, several reports have investigated the use of OMVs as antibiotic delivery carriers or as active antibacterial agents [15,25,26], showing the potential role of OMVs in antibacterial therapy (Figure 4I). Traditional antibiotic therapy has faced many challenges due to the emerging antibiotic resistance in bacteria that has resulted in treatment failure and is becoming a serious threat to human health. The earliest observation of the bactericidal effect of OMVs was reported in 1996, when OMVs from *P. aeruginosa* were found to contain peptidoglycan hydrolases (autolysins) [21]. Autolysins are intracellular bacteriolytic peptidoglycan hydrolases that are commonly found in bacteria and play major roles in various essential functions such as protein transport, cell division, and peptidoglycan recycling [114]. OMVs have the ability to transport autolysins into other competitor bacteria (Gram-positive and Gram-negative) and negatively impact them by causing disintegration through hydrolyzing their peptidoglycan. Moreover, OMVs from *Myxococcus xanthus* (*M. xanthus*) were reported to have many types of enzymes such as phosphatases, hydrolases, and proteases with bactericidal activity against *E. coli* [115]. The antibacterial activity of these OMVs against *E. coli* was improved when the fusogenic enzyme (glyceraldehyde-3-phosphate dehydrogenase) was present, which facilitated the fusion-based interaction between *M. xanthus* OMVs with the targeted cells. Similarly, OMVs from *Lysobacter capsici* (*L. capsici*) were found to have a bactericidal effect due to their bacteriolytic enzymes [116] and OMVs from *P. aeruginosa* were found to have significant antimicrobial activity against *Staphylococcus epidermidis* (*S. epidermidis*) due to the presence of quinolines within the OMVs [23]. In this regard, the interbacterial antagonism between at least two bacteria can be explored in order to use their OMVs as antibacterial agents. 

In addition to their natural antibacterial activity, OMVs can be also used as antibiotic delivery carriers due to their efficient targeting capacity, drug loading, cargo protection, prolonged circulation time, etc. [107,117]. It is reported that the targeting capacity, pharmacokinetics properties, and chemical stability of antibiotics can be enhanced when loaded inside the OMVs [25]. Gentamicin-loaded OMVs were found to have a strong bactericidal effect against the gentamicin-resistant *P. aeruginosa* [21]. Similarly, the gentamicin-loaded OMVs from *Buttiauxella agrestis* (*B. agrestis*) exhibited a strong bactericidal effect against their parental cells as well as against *E. coli* and *P. aeruginosa* [118]. Interestingly, the bactericidal effect of gentamicin-loaded OMVs was stronger against *B. agrestis* (parental cell) and other bacterial species of *Buttiauxella* spp. than those of *E. coli* and *P. aeruginosa*, suggesting a bacterial species specificity. Despite the multiple advantages that OMVs can offer as antibiotic delivery vesicles, only a few reports have been published so far, indicating the need for further investigation. 

## 3. *Helicobacter pylori* (*H. pylori*)

*H. pylori* is a widespread gastric Gram-negative bacterium infecting half of people worldwide, and in some countries, over 70% of the population [119,120,121]. Its transmission occurs via fecal–oral, gastric–oral, or oral–oral routes [122,123,124], and it can also be linked to routine esophagogastroduodenoscopies as a consequence of ineffective disinfecting procedures [125,126,127,128].

Although the infection could be asymptomatic in some patients, it may progress into severe gastric diseases and disorders [129]. Indeed, *H. pylori* infection is often associated with several gastrointestinal diseases such as peptic ulcer, gastritis, gastric lymphoma, and gastric carcinoma, and is linked with extragastric diseases including vitamin B12 deficiency, idiopathic thrombocytopenic purpura, and idiopathic iron deficiency anemia, as well as metabolic, cardiovascular, colorectal, and neurological disorders [130,131,132]. In addition, *H. pylori* infection has been associated with gastric cancer, which is reported to be the third most common related type of cancer and the second resulting factor for cancer-related death [133,134]. 

*H. pylori* infection and its associated diseases remain persistent and difficult to treat, reaching alarming levels globally. This could be attributed to *H. pylori* resistance against antibiotics such as levofloxacin, clarithromycin, and metronidazole often used to treat the infection [135]. In addition, the commonly used proton pump inhibitor (PPI)-based triple therapy that includes a PPI plus two antibiotics showed limited success, and therefore, it is unacceptable for *H. pylori* therapy (Table 1). A number of treatment regimens were proposed, including triple therapy, quadruple therapy, sequential therapy, and probiotics therapy, but the treatment choice is highly dependent on the availability of susceptibility testing as well as the effectiveness of local empiric therapy (Table 1) [136]. Therefore, it is essential to find other more effective measures to overcome *H. pylori* infection and its associated diseases.

### H. pylori Components and Their Potential in Immune Modulation

*H. pylori* infection induces complex immune responses in the host that include innate and adaptive mechanisms (Table 2) [145,146,147]. Gastric epithelial cells are a critical component of the innate immune response against *H. pylori*, interfacing with it before the establishment of the infection [148,149]. Once *H. pylori* come in contact with the gastric epithelial cells, pathogen-associated molecular patterns (PAMPs) will be activated, including toll-like receptors (TLRs) and nucleotide oligomerization domain 1 (NOD1). Gastric epithelial cells express TLRs (e.g., TLR1, TLR4, and TLR5), which interact with different *H. pylori* components such as neutrophil-activating protein (Nap), flagellin, lipopolysaccharide (LPS), lipoproteins, and lipoteichoic acid [150,151,152]. The TLRs are critical in inducing the expression of antibacterial and proinflammatory factors [150]. 

When *H. pylori* infection occurs, neutrophil cells could be observed, and their presence could also be detected in chronic infections of *H. pylori* during adulthood. Their presence is attributed to cytokines induced by *H. pylori*, which result in neutrophils activation and regulation of their movement (e.g., growth-related oncogene (GRO)-α and IL-8) [153]. 

The specific immune responses toward *H. pylori* that are associated with its bacterial cell components have been discussed in in vivo and in vitro studies, and are summarized in Table 2.
ijms-24-08542-t002_Table 2Table 2Various *H. pylori* components with some of their immunogenic activities.*H. pylori* Immunogenic ComponentImmune ResponseRefs.Vacuolating cytotoxin A(VacA)-Monocytes U937, colon epithelial cells DLD-1, and gastrointestinal epithelial cells MKN1: IL-8 release.-T cells: inhibition of IL-2 production.-Regulates T cell activation.-Inhibits primary T cell proliferation.-Interferes with B lymphocyte antigen presentation.-Invokes the secretion of circulating VacA antibodies.-Bone-marrow-derived mast cells: produce proinflammatory cytokines, IL-13, IL-10, IL-6, IL-1β, macrophage-inflammatory protein-1α, and TNF-α.[154,155,156,157,158,159]Cytotoxin-associated gene A (CagA)-Stimulates IL-12, IL-1β, COX-2, and IL-8 release.-Promotes Treg cell differentiation.-Gastric epithelial cells: regulate autophagy pathways.[160,161,162,163,164,165]Urease-Gastric epithelial cells: stimulate IL-6, TNF-α, and IL-8 secretion.-Induces Th2 cell response.[166,167,168,169]Flagellum-Induces antibody responses.[170,171,172]Catalase-Elicits inflammatory response.[173]Superoxidase dismutase (Sod)-Inhibits pro-inflammatory cytokine production.-Induces macrophage activation.[174]Lipopolysaccharide(LPS)-Interferes with innate and adaptive immune cell activities.-Increases TNF-α and IL-10, IFN-γ, IL-2, and IgG2a serum levels.-Promotes a Th1-type immune response.-Affects adaptive T lymphocyte response.-Promotes chronic inflammation.-Activates neutrophils.-Stimulates monocyte transendothelial migration and monocyte inflammatory responses.-Human monocytes: induce the release of IL-8, monocyte chemotactic protein 1 (MCP-1), and epithelial neutrophil-activating peptide 78 (ENA-78).-Heparinized human peripheral whole blood (HPWB) cells: promote IL-18 and IL-12 production.[175,176,177,178,179,180,181,182,183,184]Blood group antigen-binding adhesin (BabA)-Stimulates IL-33 expression.-Stimulates granulocyte infiltration.-Promotes IL-8 release.-Enhances gastric inflammation.[185,186]Sialic acid-binding adhesin (SabA)-Stimulates neutrophil infiltration and activation.[187]Outer inflammatory protein A (OipA)-Induces inflammation.-Promotes neutrophil infiltration.-Gastric cancer cells: stimulate interferon regulatory factors 1.-Induces the proinflammatory cytokines production such as IL-17, IL-11, IL-8, IL-6, IL-1, TNF-α, CC chemokine ligand 5, and matrix metalloproteinase-1.-Inhibits the dendritic cells’ maturation.[188,189,190,191,192]Duodenal ulcer promoting gene A (DupA)-Enhances neutrophil infiltration.-Stimulates IL-8 and IL-12 production (IL12-p70 and IL-12p40).[193,194,195,196]Adherence-associated lipoprotein A and B (AlpA/AlpB)-Gastric epithelial cells: regulate cytokines and pro-inflammatory factor release such as IL-6 and IL-8.[197]Induced by contact with epithelium gene A (IceA)-Promotes the production of the proinflammatory cytokines IL-8, IL-1, and IL-6.-Induces lymphocytic and granulocytic infiltration.[198,199,200]Cholesteryl α-glucosyltransferase (αCgT)-Regulates responses from the IFN-γ and IL-4 pathways.-CD4+ T cell response regulation.-Stimulates IL-8 production.-Inhibits IL-22 and IL-6 signaling pathways.-Macrophages: modulate autophagy.-Arrests macrophages’ phagosome maturation.[201,202,203,204,205]γ-glutamyl-transpeptidase (Ggt)-Induces IL-8, IFN-γ, IL-6, and inducible nitric oxide synthase production.-Induces immune tolerance by inhibiting dendritic cell differentiation and T cell-mediated immunity.-Infiltrates CD8+ cells into the gastric mucosa.[206,207,208]Neutrophil-activating protein (Nap)-Stimulates infiltration of polymorphonuclear granulocytes and monocytes.-Stimulates the release of IL-8, IL-12, IL-23, IFN-γ, and IL-6.-Induces activation of neutrophils, mast cells, and monocytes.-Facilitates Th1 immune responses and at the same time inhibits Th2 responses[209,210,211,212,213,214,215,216,217]Heat shock protein 60 (Hsp60)-Gastric epithelial cells: TLR activation.-Promotes the upregulations of cytokines such as TGF-β, TNF-α, IFN-γ, IL-10, IL-8, and IL-1a.[218,219,220]


*H. pylori* contributes to activate both cellular and humoral immune cells (e.g., dendritic cells, B and T cells) and activates both local and systemic immunological reactions, favoring the secretion of IgG, IgM, and IgA [221,222]. Moreover, inflammatory reactions resulting from immunity caused by polymorphonuclear monocytes and leukocytes produce several cytokines, such as IL-6, IL-1β, IL-8, and TNF-α. Importantly, the secretion of IL-12 from dendritic cells and macrophages leads to Th1 cell activation and then results in cytokines production (e.g., IFN-γ). When *H. pylori* causes macrophage stimulation, it induces the production of the pro-inflammatory molecule nitric oxide by nitric oxide synthase 2 (NOS2) and inducible nitric oxide synthase (iNOS). T cells are considered the most important components of the immunological reaction during *H. pylori* infection [223]. It is well established that cytokines such as IFN-γ and/or IL-10 are produced with the activation of Th1 and Th2 cells by *H. pylori* infection, respectively. Initially, all T cells are in a resting state known as Th0, which appear in a non-polarized phenotype; however, they are able to differentiate into effective T helper cells [224], and this can produce predominant cytokines (e.g., Type 1 or Type 2). Nevertheless, polarized Th cells show an important role in overcoming *H. pylori* infections, which leads to the production of *H. pylori*-specific antibodies. Thus, T helper cell responses were demonstrated as more significant for *H. pylori* attenuation [225,226]. In the microenvironment, pathogen components, as well as the genetic factors of the host, fluctuate the differentiation of Th1 or Th2 through leader cytokines’ promotion. For type Th1 development, IL-8, IL-12, and IFNs are powerful stimuli, while IL-4 promotes type Th2 immune response. In this regard, the type of T helper cell response against *H. pylori* is highly dependent and might vary according to the *H. pylori* components involved [227]. For instance, LPS was found to induce Th1 immune response, while urease subunit B (UreB) induced Th2 cell response [169,228]. In vivo and in vitro studies are summarized in Table 2.

## 4. OMVs as Immune-Modulating Agents against *H. pylori* Infections

Immune modulation in biomedical applications represents an attractive therapy that uses the patient’s immune system to ensure disease suppression, remission, and elimination [229], which might be an effective strategy to fight diseases. As described above, *H. pylori* contain various components, with a role in maintaining bacterial cell function or/and pathogenicity, that could act as immune-modulating candidates [230], and their deep investigation will likely allow the development of innovative vaccines and therapeutic targets [231,232]. In this scenario, OMVs contain various components from the parental cell, and thus are able to induce the same immune response of the bacterial cell, and may represent promising tools [78]. Although OMVs from *H. pylori* have been explored to this aim both in vitro and in vivo (Table 3), to the best of our knowledge, no promising clinical trial outcomes have been reported so far. Indeed, many factors should be considered in order to fully explore OMVs as immune-modulating agents, such as their potential cytotoxicity or lack of efficacy in immune regulation.

### 4.1. Using OMVs Isolated from Standard or Manipulated Growth Conditions or from a Specific Growth Stage

Previous reports have indicated that Th2 immune reaction is necessary for immunological protection against *H. pylori* infections. Thus, a viable *H. pylori* immune-modulating agent necessitates a qualitative shift in the T cell response balanced to the Th2 response [18,225,226,240,241,242]. Liu et al. [18] observed that the oral administration of OMVs from the gerbil-adapted *H. pylori* strain 7.13 elicited Th2 immune response, whereas the groups treated with the whole-cell antigens as well as the whole-cell antigens plus Cholera toxin had distinct outcomes, with a balance of Th1 and Th2 responses. 

Four distinct populations of OMVs were isolated from *H. pylori* SS1, induced by different growth additives. The biological properties and contents of all four OMVs were different [233], but they all elicited a Th2 immune response and released anti-inflammatory cytokines. OMV populations may differ in terms of the composition obtained from their parental cell based on the change in external factors (e.g., temperature, growth media) or based on the change in the bacterial strain. 

Growing bacteria under different conditions that mimic the conditions of the human gastrointestinal tract were found to produce OMVs with different cytotoxic activity compared to the bacteria that were grown under standard laboratory conditions [234]. Furthermore, these gastrointestinal-tract-mimicking conditions were found to upregulate the production of OMVs. In addition, the manipulation of growth conditions might be a good strategy to find suitable OMVs for immune modulation [233]. The growth stage was also reported to affect OMV properties. OMVs obtained from the pre-stationary phase were suggested to be better suited for vaccine research studies [235]. Thus, a change in OMV composition could lead to a different biological activity. 

### 4.2. Using OMVs from Bacterial Strains That Contain Nontoxigenic Virulence Factor Genotypes (e.g., CagA, VacA, DupA) or That Lack Certain Virulence Factors (e.g., CagA-Negative H. pylori Strains, DupA-Negative H. pylori Strains)

Various concerns have been raised about OMV safety and potential to modulate immune responses. It is important to select the OMV population that elicits an immunological response while having no cytotoxic effect on the cells and the host [17,233]. The immunogenicity of any OMV population will be determined based on its contents, which might include several immunogenic components that can induce immune responses. However, some of these immunogenic contents are virulence factors that affect the host cells negatively. For instance, VacA was found to stimulate immune responses; however, its cytotoxic effect limits its use in vaccine development [243,244]. All *H. pylori* strains have VacA, which is a critical virulence factor that is involved in bacterial survival and colonization in gastric epithelial cells. However, VacA can be present in various isoforms in different *H. pylori* strains that are linked to different degrees of gastrointestinal disease severity as well as differential cell toxicity [157,245]. Thus, only some *H. pylori* strains produce the pathogenic and toxigenic VacA [157,236,237]. For instance, *H. pylori* containing VacA type s1/m1 showed high cytotoxic activity compared to type s1/m2-expressing strains, while none of the type s2/m2 showed cytotoxic activity. These observations could be beneficial in biomedical applications allowing the use of VacA with no cytotoxic activity. For instance, type s2/m2 and type s1/m1 VacA toxins are identical in about 75% of amino acid sequences, which could be beneficial in the area of developing safe VacA-based immunomodulating agents. Thus, the OMVs derived from these *H. pylori* strains (e.g., s2/m2) could be useful and good candidates for immune-modulating applications. 

Similar approaches could be attempted with other *H. pylori* virulence factors/proapoptotic factors, which include LPS, CagA, Urease, γ-glutamyl transpeptidase, and FasL [176,246,247,248,249,250]. For instance, *H. pylori* CagA is highly immunogenic, causing a Th1-polarized immune response [251]. Nevertheless, similar to VacA, it can be present in different isoforms, some of which lack cytotoxic effects. Similarly, OMVs derived from these strains can be beneficial for immune modulation.

Duodenal ulcer promoter A (DupA) protein is one of the virulence factors of *H. pylori* [189]. DupA-positive *H. pylori* strains cause high-level gastric inflammation [194] and the development of duodenal ulcers [252]. However, in Western populations, the presence of DupA in *H. pylori* strains was not associated with duodenal ulceration [253,254]. Indeed, some DupA isoforms lacking a negative impact on health were identified, supporting the use of DupA as a potential immunogen for *H. pylori* treatment [196,254]. OMVs derived from *H. pylori* strains that contain DupA isoforms with no negative effect can be good candidates for further investigation.

Of note, the positive/negative *H. pylori* strains of any cellular component (e.g., CagA-positive *H. pylori* strains, CagA-negative *H. pylori* strains, DupA-negative *H. pylori* strains, etc.) [155,157,238,245,253,255] could also provide a valuable understanding of how to design immunogenic agents for immune modulation to treat *H. pylori* infections. Thus, their derived OMVs can provide a good strategy to optimize their use in immune modulation.

### 4.3. Using OMVs from Probiotic or Commensal Bacteria as Antigen Carriers for the Antigens of Interest

Probiotic or commensal bacteria can be genetically engineered to express immunogenic components (e.g., VacA from *H. pylori* type s2/m2 strains) inside the OMVs or on OMV surfaces. Commensal bacterial strains are part of the human gut microbiota and their derived OMVs are considered key players in biological processes inside the intestinal mucosa [256]. Moreover, probiotic bacteria have been widely recognized to have a vital role in regulating intestinal health [257,258]. OMVs produced by probiotic or commensal bacteria have been linked to various beneficial effects on the host, such as maintaining intestinal homeostasis and signaling processes [256,259]. Therefore, they are safe and can be considered good antigen carriers for the antigen of interest. This modification can be applied by loading the desired antigen (e.g., VacA) directly inside the OMV lumen, by linking the desired antigen on the OMV surfaces, or by genetically engineering the commensal/probiotic bacteria to express the desired antigen in their produced OMVs [15]. The use of probiotics during the *H. pylori* treatment process has been reported to increase the eradication rate as well as to reduce treatment side effects [144]. However, probiotic treatment was not recommended as a single strategy against *H. pylori*. Probiotics can assist in *H. pylori* eradication through various mechanisms such as producing antibacterial substances, competing with *H. pylori* for adhesion receptors, and stabilizing gut mucosal barriers [142]. Thus, can be beneficial to include them in therapy. Engineering probiotic bacteria to contain *H. pylori* components can facilitate their use as immune-modulating agents for *H. pylori* infections. On the other hand, these engineered probiotic bacteria can be employed to produce OMVs that can be further used in immune modulation.

A non-toxigenic strain of *Vibrio cholerae* (*V. cholerae*) was engineered to express *H. pylori* adhesin A (HpaA). The oral immunization of mice using the inactivated *V. cholerae* expressing HpaA showed high anti-HpaA responses in the serum [260]. This demonstrates that non-toxigenic bacterial strains can be genetically engineered to allow the surface expression of *H. pylori* components, and therefore, these recombinant non-toxigenic bacterial strains can be considered as oral inactivated *H. pylori* vaccines. Moreover, OMVs from these recombinant non-toxigenic bacterial strains can be investigated further to test their potential as immune modulatory agents.

## 5. Conclusions

OMVs have gained recognition as viable candidates for a wide range of biomedical applications. They play important roles in different bacterial biological processes such as bacterial virulence and cellular crosstalk. Moreover, they have desirable properties such as their ability to induce immune responses. *H. pylori* causes several serious gastrointestinal burdens. Due to the persistence of *H. pylori* infections, antibiotic resistance, and low treatment success, as well as low effectiveness of the current treatment/prevention regimens, it is important to explore other strategies to fight *H. pylori* infections and their associated gastrointestinal diseases, and OMV-based immune modulation can be an attractive approach. OMVs have several abilities that enable them to be good candidates in immune modulation, such as their resemblance to the parental cell as well as their ability to induce immune responses; therefore, they can act as effective immunomodulating agents. Overcoming the possible limitations of using OMVs, such as their possible toxicity, can be explored to enhance their safety and avoid any possible adverse effects. In this regard, different strategies have been and could be considered in the future, such as using the wild-type OMVs from specific *H. pylori* strains, or engineering OMVs to achieve certain desirable characteristics such as reducing OMV toxicity or enhancing their immunomodulatory effect. Moreover, enhancing the current OMV isolation and loading techniques as well as improving their yield can be a good strategy to facilitate obtaining the optimum measures for accelerating using OMVs in various medical fields. Overall, the acquired knowledge and ongoing advances in this research field can allow us to broaden our understanding of how to fully harness OMVs to be used as immunomodulating agents to fight several pathogens that cause serious diseases. 

## Figures and Tables

**Figure 1 ijms-24-08542-f001:**
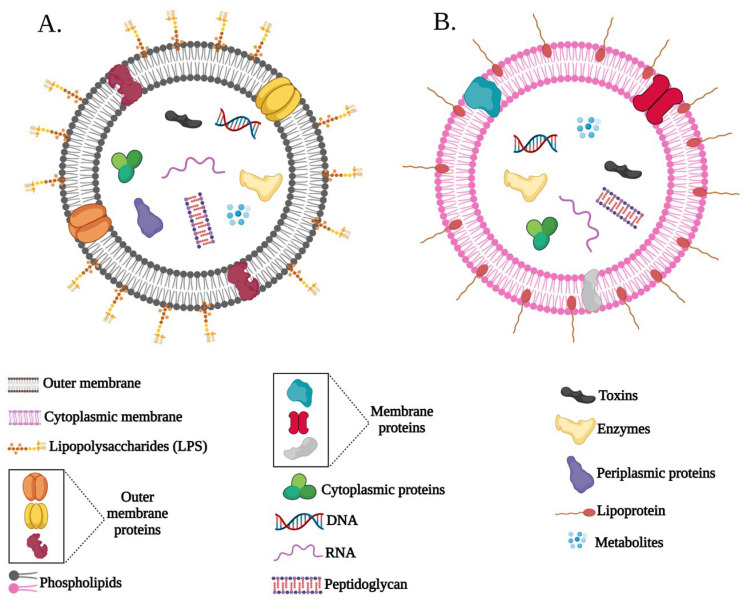
A graphical overview of extracellular vesicles from Gram-negative bacteria (outer membrane vesicles, OMVs) and Gram-positive bacteria (membrane vesicles, MVs). (**A**) OMV is constituted by a continuous lipid bilayer originating from the outer membrane and composed of various cytoplasmic and outer membrane proteins, toxins, enzymes, nucleic acids, peptidoglycans, and biomolecules derived from their parental bacterial cell. (**B**) MV is composed of a continuous lipid bilayer originating from the cytoplasmic membrane, and contains different components such as membrane proteins, cytoplasmic proteins, enzymes, toxins, and nucleic acids.

**Figure 2 ijms-24-08542-f002:**
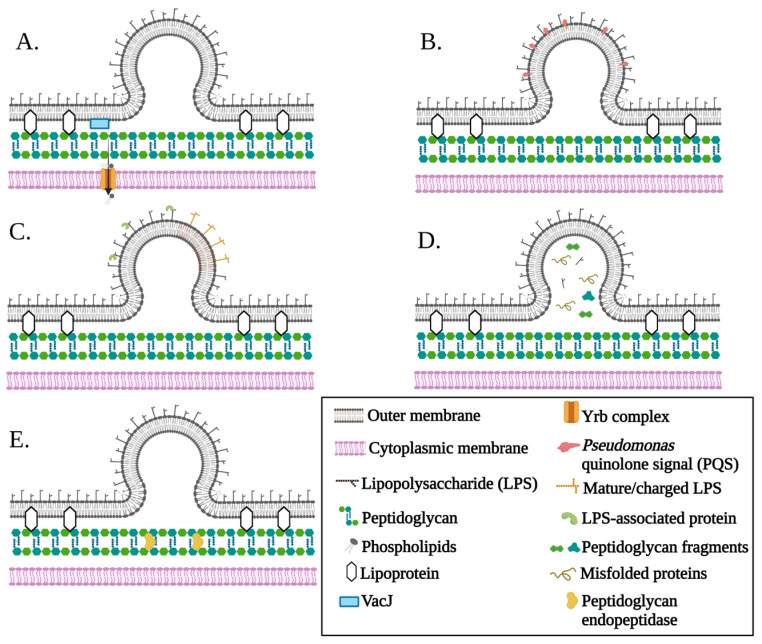
Current reported OMV biogenesis models. (**A**) VacJ/Yrb ABC transporter downregulation. Vesiculation is promoted via VacJ/Yrb transporter downregulation that causes phospholipid accumulation inside the outer membrane leaflet. (**B**) The insertion of molecules that trigger the outward bulging of the outer membrane (e.g., *Pseudomonas aeruginosa* (*P. aeruginosa*) quinolone signal, PQS). PQS insertion in the outer leaflet of the outer membrane promotes the curvature of the membrane and causes OMV formation. (**C**) Enrichment of specific components/molecules in some parts of the outer membrane. Various types of bacterial components such as lipopolysaccharides (LPS), LPS-associated molecules, and phospholipids can enrich certain parts of the outer membrane, which leads to bulging outward, forming OMV. This phenomenon occurs due to the unique structure or charges of these components. (**D**) Envelope components’ accumulation. The accumulation of various components such as misfolded proteins, LPS, or peptidoglycan fragments creates a pressure that induces the formation of OMVs. This stress pressure allows the outward bulging of the outer membrane and ultimately releases the OMVs at the areas where accumulation occurred. (**E**) Peptidoglycan–lipoprotein crosslink disruption. Enzymes controlling peptidoglycan synthesis and breakdown (e.g., peptidoglycan endopeptidases) regulate the envelope’s ability to create peptidoglycan–lipoprotein crosslinks. Under this situation, the outer membrane growth rate would be faster than the growth rate of the cell wall underneath, which permits the outer membrane to curve and consequently release the OMVs.

**Figure 3 ijms-24-08542-f003:**
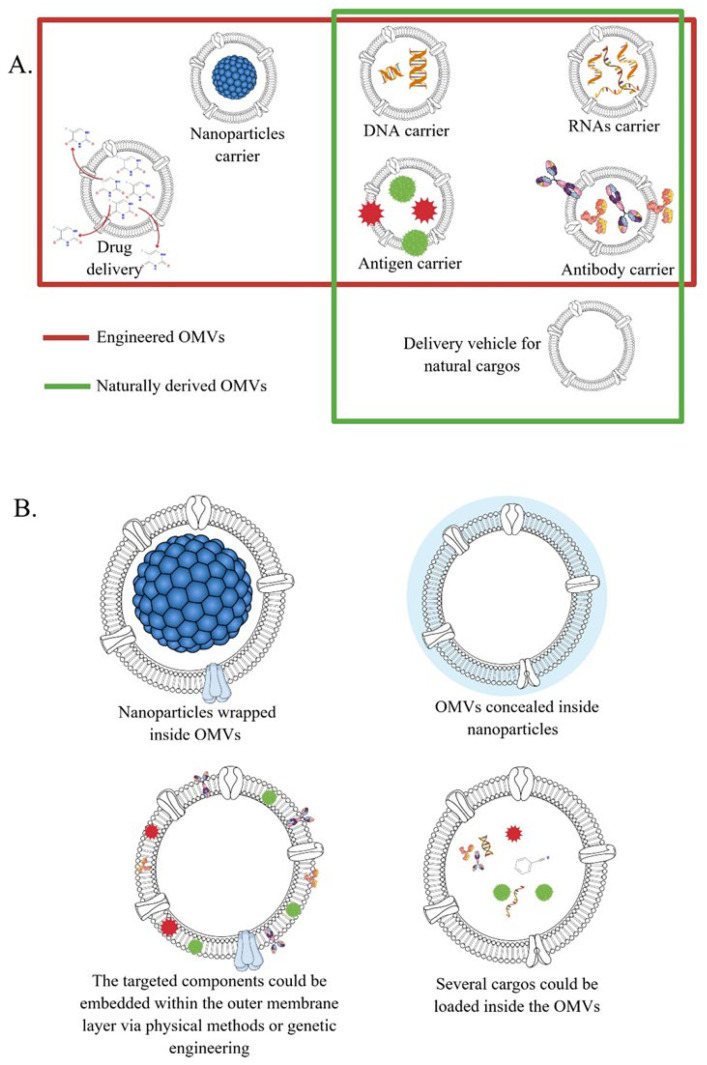
Potential use of natural and engineered OMVs. (**A**) Natural OMVs can be used as DNA, RNA, antigen, and antibody carriers, or as delivery vehicles for natural cargos; meanwhile, modified OMVs can be used as nanoparticle, DNA, RNA, antigen, and antibody carriers, or as drug delivery vehicles. (**B**) OMVs can be modified by loading cargos inside the OMV lumen, warping nanoparticles inside the OMVs, concealing OMVs inside nanoparticles, or embedding the desired components within the outer membrane layer.

**Figure 4 ijms-24-08542-f004:**
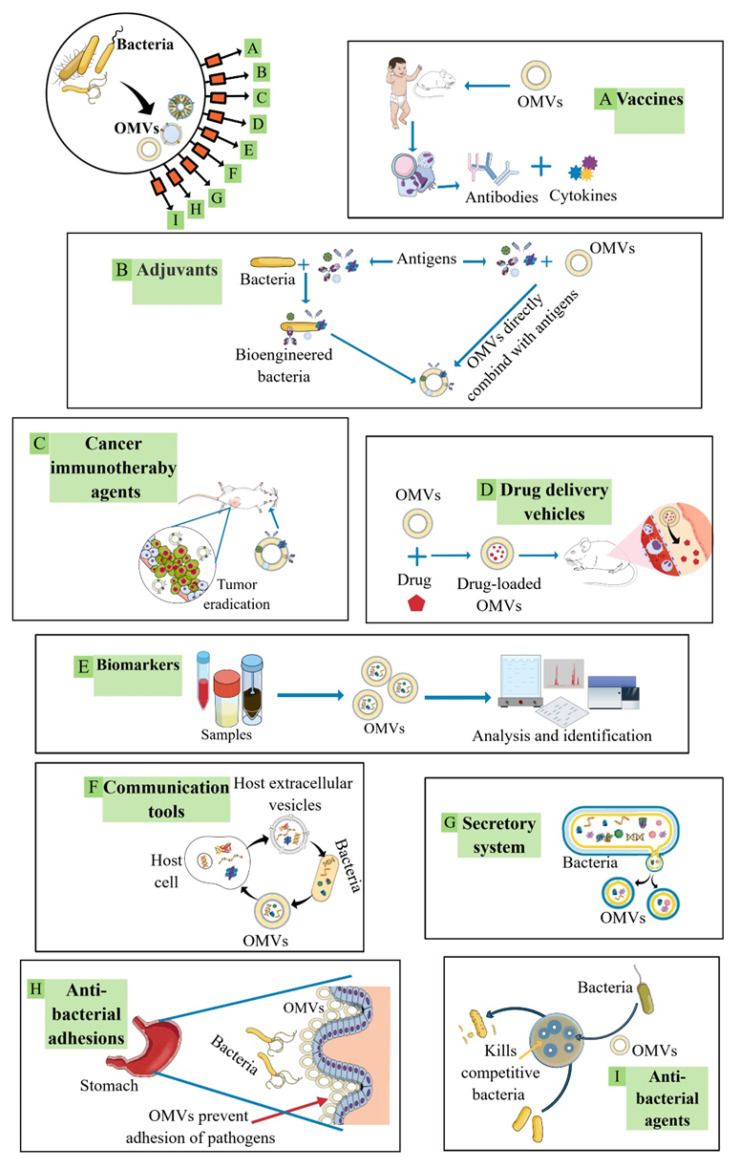
Biomedical applications of OMVs: (**A**) OMVs can be used as vaccines against pathogenic bacteria to induce cellular and humoral immune responses after immunization of humans and animals. (**B**) OMVs can be used as adjuvants that enhance the immune responses against an antigen. This can be achieved by mixing the OMVs with the antigen in the vaccine preparations. (**C**) OMVs can be used as cancer immunotherapy agents in order to eradicate tumor tissues by inducing the required immune response against cancer cells, or by acting as nanocarriers for loading chemotherapeutic agents/drugs. (**D**) OMVs can serve as a delivery system that can transport their cargo to other cells and/or microenvironments (e.g., a vehicle for drugs). (**E**) OMVs can act as ideal biomarkers for all Gram-negative bacteria or as biomarkers to differentiate between bacterial species. (**F**) OMVs can serve as communication tools by transporting signaling molecules between cells. (**G**) OMVs can act as a secretory system to disseminate bacterial products to their environment or targeted locations. (**H**) OMVs can act as anti-adhesion agents to interfere with the adhesion of pathogenic bacteria by binding competitively with the targeted host cells and subsequently preventing bacterial infection. (**I**) OMVs can be used in antibacterial therapy as active antibacterial agents or by loading antibiotics inside the OMVs.

**Table 1 ijms-24-08542-t001:** Commonly used methods to treat *Helicobacter pylori* (*H. pylori*) infection.

Method	Limitations	Refs.
Antibiotic treatment	- Antibiotic resistance is the main factor for the eradication treatment failure.	[135,136,137,138]
Triple therapy: Treatment with proton-pump inhibitor (PPI), amoxicillin, and a third drug (e.g., levofloxacin or clarithromycin).	- Antibiotic resistance. - The treatment efficacy has decreased continuously over the years. - Affected by the colonization density of the bacterium in the gastric mucosa. - Affected by host factors such as excess secretion of gastric acid, diabetes, gastroduodenal diseases, obesity, etc.	[138,139]
Quadruple therapy: Treatment with PPI, bismuth, tetracycline, and metronidazole.	- Antibiotic resistance. - Side effects. - Low eradication rate (lower than 80%).	[138,140]
Sequential therapy: Two treatment regimens are applied; the first one (consisting of PPI and amoxicillin) will be used in the first half of the treatment duration, and another regimen (consisting of PPI, clarithromycin, and one of the nitroimidazole family antibiotics) will be used for the second half of treatment duration.	- Antibiotic resistance. - The complexity of the regimen. - Adverse effects. - Difficulty to design a suitable second-line treatment if this regimen failed.	[138,141]
Probiotics therapy: The use of bacteria that produce lactic acid such as *Lactobacillus* spp., etc., to eradicate *H. pylori* infection.	- The effect of probiotics to treat *H. pylori* is still controversial (different outcomes reported from different studies). - Only a few probiotic strains have shown significant effects, which emphasizes the importance of using the right probiotics and their proper quantity. - Not recommended as a single treatment strategy to eradicate *H. pylori*.	[142,143,144]

**Table 3 ijms-24-08542-t003:** Proposed strategies for OMVs’ use, modification, and bioengineering in immune modulation to treat *H. pylori* infections.

Strategies	Refs.
Using OMVs isolated from standard or manipulated growth conditions or from a specific growth stage	[17,18,233,234,235]
Using OMVs from bacterial strains that contain nontoxigenic virulence factor genotypes (e.g., CagA, VacA, DupA) or that lack certain virulence factors (e.g., CagA-negative *H. pylori* strains, DupA-negative *H. pylori* strains)	[155,157,194,196,236,237,238,239]
Using OMVs from probiotic or commensal bacteria as antigen carriers for the antigens of interest	[15]

## Data Availability

Not applicable.

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
