# Peer review of "Outer Membrane Vesicles (OMVs) as Biomedical Tools and Their Relevance as Immune-Modulating Agents against H. pylori Infections: Current Status and Future Prospects"

_ijms, 2023, doi:10.3390/ijms24108542_

Round 1

Reviewer 1 Report

The review is very well done. It has an excellent organization of information. The flow with which the information is arranged throughout the text leads the reader to a full understanding of the problem and the proposals offered by the author.

The information placed in the work is in accordance with the references and proves to be the most up-to-date possible.

Line 54-56: “…immune modulation, drug delivery, cancer therapy, vaccine development, and anti-bacterial treatments…”,

R > Add References

Line 152: “2.2. OMVs in biomedical applications“

R > Aiming at a better understanding by the reader of the technical aspect of OMVs, I thought it might be interesting to add how a synthetic OMV can be produced and how the insertion of specific molecules into the interior of microvesicles occurs, that is, the information directed to the technical part of the process.

L: 247-250: “The OMVs cargo conserved among bacteria can act as ideal biomarkers, meanwhile, the OMVs cargo-specific for a given bacterium can be extremely useful as a rapid differentiation and identification tool for bacterial species identification (Figure 3E) [ 65].”

R > Thinking about the aspect of OMVs as biomarkers: I believe that this part could be explored a little better. How is this "position-specific" of OMV's detected?

As the author mentions in line 433-435, the variation in the constitution of OMVs seems to be very wide. Since the constitution of OMVs can vary according to the growth conditions, strain and can also according to the size of the OMVs, its use as a biomarker seems not to be very clear.

Author Response

The review is very well done. It has an excellent organization of information. The flow with which the information is arranged throughout the text leads the reader to a full understanding of the problem and the proposals offered by the author.

The information placed in the work is in accordance with the references and proves to be the most up-to-date possible.

Reply: We thank the Reviewer for the positive comments.

Comment: Line 54-56: “…immune modulation, drug delivery, cancer therapy, vaccine development, and anti-bacterial treatments…”,

R > Add References

               Reply: We thank the Reviewer for the request and apologize for the missing references. The references have been added (Lines 53-57), references numbers 15, and 17 to 26.

Comment: Line 152: “2.2. OMVs in biomedical applications“

R > Aiming at a better understanding by the reader of the technical aspect of OMVs, I thought it might be interesting to add how a synthetic OMV can be produced and how the insertion of specific molecules into the interior of microvesicles occurs, that is, the information directed to the technical part of the process.

               Reply: We agree with the Reviewer suggestion and we added the technical aspects of OMVs synthesis, loading and modifications in the text (Lines 173-202 and 207-213).

Comment: L: 247-250: “The OMVs cargo conserved among bacteria can act as ideal biomarkers, meanwhile, the OMVs cargo-specific for a given bacterium can be extremely useful as a rapid differentiation and identification tool for bacterial species identification (Figure 3E) [ 65].”

R > Thinking about the aspect of OMVs as biomarkers: I believe that this part could be explored a little better. How is this "position-specific" of OMV's detected?

Reply: We thank the Reviewer for the comment. In the text, we discussed about species-specific components in a given bacterial species that can act as a biomarker for them. We better clarify this point by adding details on how this could be achieved (Line 302-307).

Comment: As the author mentions in lines 433-435, the variation in the constitution of OMVs seems to be very wide. Since the constitution of OMVs can vary according to the growth conditions, strain and can also according to the size of the OMVs, its use as a biomarker seems not to be very clear.

Reply: We apologize for the lack of clarity in this paragraph. In this section we described  the variations of OMVs occurring in vitro when growth conditions such as temperature can be manipulated. On the contrary, the use of OMVs as biomarkers refers to their isolation in vivo such as in biological fluids as now stated on line 298. We hope to have successfully clarified this point.

Reviewer 2 Report

Comments for

Outer membrane vesicles (OMVs) as biomedical tools and their relevance as immune-modulating agents against H. pylori infections: current status and future prospects.

·       In general, this review paper was well thought out, describing the biogenesis and applications in relation to outer membrane vesicles (OMVs). However, it is ultimately unclear what are the advantages of OMVs vis-a-vis probiotics against H. pylori infections. Probiotics are more cheaply manufactured and can also produce the OMVs in situ after ingestion, conferring additional advantages of gut colonization aside what was proposed in this review paper. Perhaps the authors could elaborate more on the proposed advantages of OMVs, especially in the case of H. pylori infections, with regards to current practices or alternatives.

·       In the discussion of the biogenesis of the OMVs, while the authors have demonstrated great detail in its descriptions, a table outlining the different biogenesis possibilities would be appreciated in order to provide the information in a more succinct yet clear manner.

·       More references could be included for the loading of therapeutics in the OMVs. Additionally, more discussion on how such loading can be done is appreciated. This in turn, can be an advantage over probiotics, where therapeutics can be loaded into the OMVs for additional advantages, such as antibiotics. However, loading of therapeutics is typically challenging and thus would warrant further discussion.

·       Some discussion on isolation of OMVs, or efforts to improve its yield, whether through better isolation or enrichment techniques would be interesting.

·       Lastly, to circle back to the first point of discussion, there seems to be little discussion on the current state of the art for H. pylori infections. Hence, there seems to be a lack of a strong push factor for not using the current state of the art treatment.

Author Response

  • Comment:    In general, this review paper was well thought out, describing the biogenesis and applications in relation to outer membrane vesicles (OMVs). However, it is ultimately unclear what are the advantages of OMVs vis-a-vis probiotics against H. pylori infections. Probiotics are more cheaply manufactured and can also produce the OMVs in situ after ingestion, conferring additional advantages of gut colonization aside what was proposed in this review paper. Perhaps the authors could elaborate more on the proposed advantages of OMVs, especially in the case of H. pylori infections, with regards to current practices or alternatives.

More references could be included for the loading of therapeutics in the OMVs. Additionally, more discussion on how such loading can be done is appreciated. This in turn, can be an advantage over probiotics, where therapeutics can be loaded into the OMVs for additional advantages, such as antibiotics. However, loading of therapeutics is typically challenging and thus would warrant further discussion.

Reply: We thank the Reviewer for the suggestions that we took in careful consideration. We added a table (Line 425) to outline the current status and limitations of H. pylori treatment including using probiotics. Regarding their use for H. pylori treatment, we implement the discussion in Lines 569-576 providing more details as requested.

As requested, we added few technical aspects of OMVs loading, synthesis, and modifications in the text with proper references (Lines 173-202 and 207-213).

We finally underlined in the conclusion paragraph (Lines 602-604) a statement that underlined the need to improve the current isolation and loading techniques as well as to enhance OMVs yield .

  • Comment: In the discussion of the biogenesis of the OMVs, while the authors have demonstrated great detail in its descriptions, a table outlining the different biogenesis possibilities would be appreciated in order to provide the information in a more succinct yet clear manner.

Reply: We thank the Reviewer for the suggestion. Since Figure 4 outlines in detail the different biogenesis possibilities for OMVs, as requested, we provided the information in a more succinct clear manner adding a supplementary table (Table S1).

  • Comment: Some discussion on isolation of OMVs, or efforts to improve its yield, whether through better isolation or enrichment techniques would be interesting.

Reply: We have included this valuable suggestion in the conclusion as a good strategy to be explored further to get the full potential of OMVs in biomedical applications (Lines 602-604).

  • Comment: Lastly, to circle back to the first point of discussion, there seems to be little discussion on the current state of the art for H. pylori infections. Hence, there seems to be a lack of a strong push factor for not using the current state of the art treatment.

Reply: Thank you for your comment. A table (Line 425) was added to outline the current common used methods to treat H. pylori and their limitations, which helps to understand the need to find alternative strategies such as using OMVs.

Reviewer 3 Report

OMVs have received increasing attention as potential candidates for a wide variety of biomedical applications. They play important roles in different bacterial biological processes such as bacterial virulence and cellular crosstalk.

Moreover, they have desirable properties such as their ability to induce immune responses. H. pylori cause several serious gastrointestinal burdens. Due to the persistence of H. pylori infections, antibiotics resistance, and low treatment success as well as low effectiveness of the current treatment/prevention regimens,

It is important to explore other strategies to fight H. pylori infections and their associated gastrointestinal diseases and OMVs- based immune modulation can be a very attractive approach.

OMVs have several abilities that enable them to be good candidates in immune modulation such as their resemblance to the parental cell as well as their ability to induce immune responses, therefore, they can act as effective immunomodulating agents.

 In this regard, different strategies have been and could be considered in the future such as using the wild-type OMVs from specific H. pylori strains, or engineering OMVs to achieve certain desirable characteristics such as reducing OMVs toxicity or enhancing their immuno-modulatory effect.

Overall, the acquired knowledge and ongoing advances in this research field can allow to broaden our understanding of how to fully harness OMVs to be used as immuno-modulating agents to fight several pathogens that cause serious diseases.

Author Response

Reply: Thank you so much for this valuable and encouraging feedback.

Reviewer 4 Report

The manuscript reviews the OMVs characteristics and the molecules responsible for H. pylori's immune response. It’s well organized, internally consistent, and full of examples. However, there are little problems want to discuss with authors:

 (1)  Both the contents of “Outer membrane vesicles (OMVs)” and “against H. pylori” are little more. When posting content to be concise, direct access to the theme of the text to highlight key content out, not too many introductions. Now I need too more time to link with OMVs and H. pylori.

(2) The molecular mechanisms of immune regulation could be involved in part 3 after the Table 1.

Minor editing of English language required

Author Response

The manuscript reviews the OMVs characteristics and the molecules responsible for H. pylori's immune response. It’s well organized, internally consistent, and full of examples. However, there are little problems want to discuss with authors:

 (1)  Both the contents of “Outer membrane vesicles (OMVs)” and “against H. pylori” are little more. When posting content to be concise, direct access to the theme of the text to highlight key content out, not too many introductions. Now I need too more time to link with OMVs and H. pylori.

Reply: Based on Reviewer comment the whole text has been revised and fixed for more clarification. The “OMVs” term was only used in later texts, the word “against” was used moderately and substituted with other similar words when needed.

(2) The molecular mechanisms of immune regulation could be involved in part 3 after the Table 1.

Reply: We thank the Reviewer for the comment. According to the current reported findings, brief information about the significant immune responses against H. pylori and how they can vary according to H. pylori components has been added after the Table 2. Moreover, the reported immune responses for each H. pylori components were highlighted in Table 2 for more clarification.

Comments on the Quality of English Language

Minor editing of English language required.

Reply: Thank you for the comment. The whole text was revised and edited accordingly.

Reviewer 5 Report

This study is interesting with clinical significance. H. pylori infects half of the people worldwide. The authors put forward a new point of view to solve this problem via OMVs. The followings are some minor comments to the authors.

Comments:

1. What are the advantages of OMVs as therapeutic vehicles compared with other forms vehicles, such as lipidosome, exosome? I suggest the authors state that in 2.2. OMVs in biomedical applications.

2. Why is there no peptidoglycan inside MV in Figure 1B?  

3.The conclusion can be improved. I suggest the authors state the potential limitations of OMVs as immune-modulating agents against H. pylori infections.

Author Response

Comments:

  1. What are the advantages of OMVs as therapeutic vehicles compared with other forms vehicles, such as lipidosome, exosome? I suggest the authors state that in ‘2.2. OMVs in biomedical applications’.

Reply: We thank the Reviewer for the valuable suggestion. The advantages of using OMVs as therapeutic vesicles compared to other extracellular vesicles were highlighted in the text as suggested (Lines 154-163).

  1. Why is there no peptidoglycan inside MV in Figure 1B?

Reply: We thank the Reviewer for the comment, indeed peptidoglycans can be present and have been added inside MV in Figure 1B.

  1. The conclusion can be improved. I suggest the authors state the potential limitations of OMVs as immune-modulating agents against H. pylori infections.

Reply: Following the Reviewer request the conclusion was improved to highlight the importance of overcoming any possible limitations of using OMVs, such as their possible toxicity, to enhance their safety and avoid any possible adverse effects (Lines 597-602). Moreover, the current OMVs isolation, loading, as well as their yield, was proposed to deserve more investigation to overcome any limitations of using OMVs for biomedical applications (Lines 602-604).